# Screen-Printed Structures from a Highly Conductive Mildly Oxidized Graphene Suspension for Flexible Electronics

**DOI:** 10.3390/ma15031256

**Published:** 2022-02-08

**Authors:** Fedora Vasileva, Vasiliy Popov, Irina Antonova, Svetlana Smagulova

**Affiliations:** 1“Graphene Nanotechnology” Laboratory, Physical-Technical Institute, M.K. Ammosov North-Eastern Federal University, 677000 Yakutsk, Russia; fd.vasileva@s-vfu.ru (F.V.); vi.popov@s-vfu.ru (V.P.); 2Laboratory of Physics and Technology of Three-Dimensional Nanostructures, Rzhanov Institute of Semiconductor Physics SB RAS, 630090 Novosibirsk, Russia; antonova@isp.nsc.ru

**Keywords:** mildly oxidized graphene, screen printing, test supercapacitor, humidity sensor, film flexibility

## Abstract

In this study, the screen-printed flexible humidity sensor and supercapacitor structures from a suspension of mildly oxidized graphene (MOG) was obtained. MOG suspension with a low atomic oxygen content (~20%) was synthesized by electrochemical exfoliation of natural graphite in an aqueous solution of ammonium sulfate. MOG films (average thickness 5 μm) with a surface resistance of 10^2^–10^3^ kΩ/sq were obtained by screen printing on a flexible substrate. The thermal reduction of MOG films at 200 °C reduced the surface resistance to 1.5 kΩ/sq. The laser reduction with a 474 nm and 200 mW solid-state laser reduced the surface resistance to ~0.065 kΩ/sq. Various structures were screen-printed on a flexible substrate for a variety of flexible electronics applications. The structures representing a flat supercapacitor had an average specific capacitance of ~6 μF/cm^2^. The tensile deformations occurring during bending reduced the capacitance by 40% at a bending radius of 2 mm. Humidity sensing structures with sensitivity of 9% were obtained.

## 1. Introduction

Currently, the development of technologies for producing new two-dimensional (2D) materials and their application in the field of flexible electronics is an important scientific and practical task. In particular, the use of graphene-based materials in printed electronics allows obtaining flexible electronic circuits, various sensors, supercapacitors, etc. Screen-printing is one of the simplest and fastest printing methods, since it does not require the use of expensive equipment, compatible with a wide range of inks and substrates and is highly scalable [1].

Due to its high specific surface area, electrical conductivity and unique mechanical properties, reduced graphene oxide (rGO), is often used as flexible conducting electrodes of supercapacitors [2,3,4,5,6]. In addition, due to the presence of diverse functional groups, graphene oxide (GO) can be used as a base material for various sensors. Various studies have shown that GO is highly sensitive to the presence of water molecules and is a promising material for humidity sensors [7,8,9]. The reduction of GO does not lead to the complete removal of functional groups, which is an obstacle for obtaining good conductivity in the GO films. Thermal annealing at high temperatures of 300–450 °C is the most commonly used method for strong reduction of GO. However, such temperatures cause thermal degradation of most types of flexible polymer substrates at [10]. The development of technological methods for creating electronic structures from rGO on the surface of flexible polymer substrates with high conductivity is an important task.

GO has a high surface resistance up to ~10^12^ Ω/sq due to high content of oxygen groups [11]. Electrochemical exfoliation of graphite in an aqueous solution of an inorganic salt makes it possible to obtain graphene with a low content of oxygen and surface resistance of ~10^5^–10^6^ Ω/sq [12]. Unlike the Hammers method, most commonly used for synthesizing GO, electrochemical exfoliation does not require the use of strong oxidants, which are typically toxic and unsafe. In addition, it allows simultaneous exfoliation and oxidation processes. In addition, the yield of graphene by electrochemical exfoliation is comparatively high. The method allows a large amount of material (tens of grams) to be obtained in a short period of time (10–20 min) [13]. By changing the process parameters, it is possible to control the oxidation level and the ratio of various oxygen functional groups [14].

Among the various methods for the reduction of GO on flexible substrates, the laser reduction has a number of advantages. One of the most significant of them is the ability to control the degree of reduction and the design of the resulting structure [14]. The use of laser reduction of mildly oxidized graphene (MOG) makes it possible to obtain films and structures with good conductivity.

In this study, MOG structures on flexible substrates were obtained by screen printing. With the use of laser reduction, it was possible to achieve ultra-low values of surface resistance in the range of ~65–130 Ω/sq with the average film thickness of 5 μm. The printed supercapacitor structures were investigated. Discharge curves were measured and capacitance values were calculated. The effect of bending on the capacitance was studied. Printed humidity sensors were obtained and their sensitivity was analyzed.

## 2. Materials and Methods

MOG was synthesized using electrochemical exfoliation of graphite in an aqueous solution of ammonium sulfate followed by ultrasonic treatment. The method was described in detail in our previous work [12].

The study of the surface morphology of the obtained films and structures was carried out on a scanning electron microscope (SEM) (Jeol 7800F, Tokyo, Japan) using a EHT of 3 keV, work distance 8 mm. An elemental analysis based on energy dispersive X-ray spectroscopy (EDS) was performed using a standard Oxford Instruments (OINA, High Wycombe, UK) microanalytical nanoanalysis system for scanning electron microscopes. Investigations of oxygen functional groups on the surface of MOG particles were carried out on a SPECS photoelectron spectrometer (XPS) using a PHOIBOS-150-MCD-9 hemispherical analyzer and a FOCUS-500 X-ray monochromator (AlK, radiation, hν = 1486.74 eV, 200 W) (SPECS, Berlin, Germany). The binding energy scale (E_b_) was preliminarily calibrated according to the positions of the peaks of the core levels Au4f_7/2_ (84.00 eV) and Cu2p_3/2_ (932.67 eV). The samples were applied onto double-sided copper conductive tape 3M (USA). For an accurate calibration of the photoelectron lines, we used the internal standard method, which was the C1s line (E_b_ = 284.5 eV) from graphite-like carbon in the samples. In addition to the survey photoelectron spectra, narrow spectral regions C1s, N1s and O1s were recorded. The survey spectra were recorded at an analyzer transmission energy of 50 eV, individual spectral regions of 20 eV. The determination of the relative content of elements on the surface of the samples and their atomic ratios were carried out from the integrated intensities of photoelectron lines, corrected for the corresponding coefficients of atomic sensitivity. The automatic measuring system ASEC-03 (Moscow, Russia) was used to determine the electrical parameters of the films. Silver paste was used as an electric contact for the samples. The Raman spectra were measured using an INTEGRA Spectra (NT-MDT, Moscow, Russia) setup at a laser excitation wavelength of 532 nm.

The active layers of the flexible electronic devices were screen-printed using a stencil cut with a Silhouette vinyl cutter (Lindon, Utah, USA). Polyethylene terephthalate (PET) was used as a substrate. The structures were printed in the following way: the stencil was glued to a PET substrate and a MOG suspension was applied through a hole in the stencil, after which the layer was dried at room temperature. After drying, the stencil was removed, forming a MOG film with a given shape. By varying the thickness of the stencil, the volume and concentration of MOG, the required film thicknesses were obtained. Figure 1 shows the samples of printed structures. The distance between the electrodes (Figure 1a–c) was ~1 mm.

To increase the conductivity, the electrodes of the structure (Figure 1b) were reduced with a blue laser (wavelength 474 nm and power 200 mW).

By the screen-printing flexible supercapacitors and humidity sensors were obtained. For the structure of a flat supercapacitor, 1 cm × 1 cm squares were printed and reduced with a blue laser, representing its plates (Figure 1d). A separator was located between the plates, which was a dielectric layer made of a track membrane with a pore size of 0.2 μm. An electrolyte with the composition H_3_PO_4_/PVA/H_2_O was applied between the electrode [15]. The distance between the plates was ~600 μm.

To study the characteristics of supercapacitors, an electrical circuit was assembled (Figure 2a) for measuring the charging and discharging rates. The measurements were carried out in an automatic mode using a microcontroller. The installation was done under the following algorithm: the key SA2 was closed through the current-setting resistor R1 and C1 was charged for 1–5 s. Then, with a charge current from 1 to 100 mA, the SA2 key was turned off and the SA1 key was closed, while the supercapacitor C1 was discharged through the R2 load. Changes in the voltage across the supercapacitor as a function of time were recorded with an interval of 1 s.

For humidity sensors, a strip was printed with the following dimensions: length of 10 mm, width of 2 mm and thickness up to 6 μm (Figure 1e). The sensor structures were tested at a temperature of +25 °C in a humidity chamber (volume 2 dm^3^) with a distilled water evaporator.

Figure 2b shows a schematic diagram of the setup for automatic measurement of the dependence of the active resistance of samples on humidity. The setup was based on the nonequilibrium potential Wheatstone bridge. The voltage on the measuring diagonal of the bridge circuit is determined by the equation 1:(1)$$U=U_{0}\frac{R_{1}R_{x}-R_{2}R_{3}}{\left(R_{1}+R_{2}\right)\left(R_{3}+R_{x}\right)}$$
where *U*—measured voltage, *U*_0_—constant voltage, *R*_1_, *R*_2_, *R*_3_–constant resistances in Wheatstone bridge, *R_x_*—resistance of the humidity sensor.

With small changes in *R_x_*, this voltage varies linearly in proportion to the change in Rx.

This voltage was amplified by the operational amplifier DA1 and fed to the ADC input of the microcontroller (MC) DD1. The MC software (Arduino IDE 1.8.19) converted the measured voltage into resistance Rx. At the same time, the output voltage from the commercial capacitive humidity sensor DA2—HIH4000 was measured. The DA2 output voltage was linearly proportional to the relative air humidity in the humidity chamber, which contained the test sample and the sensor (the chamber is shown by a dashed rectangle in the diagram). At the end of the measurement cycle, the MC sent the humidity values and the measured resistance value of the sample through the serial port to the PC, where the samples with the same cycle time were written to a text file.

## 3. Results and Discussion

### 3.1. Structural Changes in MOG Films during Their Reduction

The study of surface morphology, composition of functional groups and electrical properties of the resulting structures was carried out. Determination of the type and ratio of functional groups in MOG films were made using the XPS method. The results are shown in Figure 3.

In the range of binding energies E_b_~280–294 eV for MOG (Figure 3a) and GO (Figure 3b), the following peaks were found: the most intense peak with a binding energy E_b_~284.5 eV corresponding to carbon-carbon bonds with sp^2^ hybridization; a peak with a binding energy E_b_~286.7 ± 0.1 corresponding to epoxy and hydroxyl groups (C-O-C, C-OH (28%)); a peak with a binding energy E_b_~288.4 ± 0.1 eV corresponding to ketone and carboxyl groups (C=O, COOH (6%)). The O/C ratio = 0.36 for MOG, while for GO O/C = 0.53.

The analysis of the spectra in the range of binding energies E_b_~394–406 eV (Figure 3c) suggests that nitrogen on the sample surface was included in the following functional groups: E_b_~399.5 eV—pyrrole nitrogen and E_b_~401.5 eV—nitrogen bound with graphite-like carbon. In the range of E_b_~528–538 eV (Figure 3d), there were peaks with E_b_ = 531.3–531.6 eV from oxygen in the composition of carbonyl or carboxyl groups, while the range of E_b_ > 532.7 eV was characterized by peaks from hydroxyl and ester groups.

To increase the conductivity of MOG films on flexible substrates, two types of reduction were used. The first MOG sample was thermally reduced at 200 °C (at temperatures above 200 °C, the PET substrate begins to deform) for 30 min. The second sample was reduced with a 474 nm, 200 mW blue laser. The surface MOG morphology is shown in Figure 4.

As shown in Figure 4a, before the reduction, the surface of the MOG film was relatively uniform. After the thermal reduction, the roughness of the film surface increased (Figure 4b). In addition, during the laser reduction (Figure 4c), the appearance of round formations up to 30 μm in size, most likely, corrugations, was observed. The appearance of corrugation in GO films was observed during the reduction of GO with hydrazine vapor in our previous works [11]. The increase in the surface roughness can be explained by the formation of gaseous products (for example, H_2_O, CO_2_, CO, or other small gas molecules), which accumulate in the interlayer spaces of the MOG. Thus, a higher rate of MOG reduction under laser irradiation (as compared to heat treatment) and, as a consequence, intense gas release leads to surface deformations and an increased porosity of the reduced mildly oxidized graphene (rMOG) films. Similar changes in the morphology of reduced GO films were observed, for example, in [16].

The degree of reduction was determined from the elemental analysis data. The EDS spectra are shown in Figure 5.

The atomic oxygen content before reduction was about ~20%, after thermal reduction it was 7% and after laser reduction, it decreased to ~1.7%. It can be concluded that the reduction of MOG films on flexible substrates with a solid-state laser has a number of advantages. The flexible substrate is not significantly heated and the MOG film absorbs almost all of the irradiation energy. High energy transfer efficiency results in efficient removal of oxygen groups and the production of films with high porosity and degree of reduction. At the same time, thermal reduction at low temperatures (up to 200 °C), which does not damage the PET substrate, showed a relatively weak reduction of the MOG film.

The Raman spectra of MOG films before and after laser reduction and thermal annealing are shown in Figure 6. The spectrum of the initial MOG film was typical for GO. Peak G (1613 cm^−1^) corresponded to vibrations of carbon bonds with sp^2^ hybridization, peak D (1345 cm^−1^) was associated with defects and sp^3^ hybridized carbon atoms, peaks 2D (2703 cm^−1^) and 2D’ (2935 cm^−1^) referred to two-phonon lattice vibrations. The reduction (both laser and thermal) led to a significant decrease in the I_D_/I_G_ ratio. In the case of laser reduction, this ratio was minimal at about 0.70. An increase in the intensity of the 2D peak (I_G_/I_2D_ < 1) and the disappearance of the 2D peak make the Raman spectrum of the samples similar to the spectrum of graphene. This change is most likely associated with the formation of thermally expanded graphene upon laser irradiation. In addition, after reduction, the G peak was observed to shift from 1613 to 1592 cm^−1^ in the case of thermal reduction and 1584 cm^−1^ in the case of laser reduction. As shown in [17,18,19], the reason for the shift is the elongation of carbon-carbon bonds under tension. The reduction introduces deformations and the magnitude of the deformations is maximum in the case of laser irradiation, which is consistent with the data in Figure 4.

### 3.2. Electrical Characteristics of rMOG Films

Measurements of the electrical characteristics of MOG films on flexible substrates were carried out before and after reduction. Before reduction, the MOG films had a surface resistance of ~10^2^–10^3^ kΩ/sq. MOG films after thermal reduction at 200 °C had surface resistance ~1.5 kΩ/sq. It is known that the removal of hydroxyl and epoxy groups occurs at these temperatures [16]. During laser reduction, the surface resistance of MOG films deposited on a PET substrate decreased to ~0.065 kΩ/sq (Figure 7) at a film thicknesses in the range of 4–6 microns.

We investigated the effect of the speed of movement of a laser beam over a surface on a decrease in the surface resistance of MOG films. The results are shown in Figure 7.

It was found that the minimum value of the surface resistance of the rMOG (0.065 kΩ/sq) can be achieved at a laser beam speed of 5 mm/s. When the laser beam speed was less than 5 mm/s, the PET substrate heated up, which led to its deformation. At a higher speed of the laser beam, the power absorbed by the film would presumably decrease and, thus, the removal of functional groups from the film surface would occur more slowly.

To determine the capacitances of the capacitor structures, the discharge curves were taken according to the diagram shown in Figure 2. The discharge curves have the form of an exponential function, therefore, to calculate the capacitance, formula 2 was used:C = τ/R,(2)
where C is the capacitance, R is the load resistance and τ is the discharge time.

The lateral structures shown in Figure 1a,c had small capacitances, within 10 pF/cm^2^, since the dielectric was the air gap and the electrodes were not reduced. For the structures shown in Figure 1b, the electrodes were reduced with a laser beam and an electrolyte was deposited between the electrodes. The dependence of the voltage on the discharge time for this structure is shown in Figure 8a. The capacity calculated from the discharge curve was ~0.6 μF/cm^2^, while the capacity of this structure before reduction and electrolyte deposition was ~4 pF/cm^2^. Laser reduction of the MOG electrode increases the porosity of the material, which leads to an increase in the electrode area and, consequently, the capacity.

The highest capacitance values were obtained for vertical supercapacitor structures consisting of square electrodes with an area of 1 cm^2^, located opposite each other. The electrodes of this structure were also reduced by the laser and an electrolyte and a separator were deposited between the electrodes. The discharge curves for this supercapacitor were measured at different bending radii. The results are shown in Figure 8b.

The calculated capacitance for the structure before bending was ~6 μF/cm^2^. The results of influence of the bending radius on the value of the supercapacitor capacity are shown in Figure 8c. It was found that the capacitance values began to change from the values of bending radii of ~6 mm (with a substrate thickness of 100 μm, the deformation is ε = 0.8%). At a maximum bending value of 2 mm (ε = 2.5%), the capacitance decreased to ~3.5 μF/cm^2^ and practically did not recover after the deformation was removed. The deformations of the electrodes arising during bending may lead to a decrease in the effective area of the electrodes due to their cracking, which results in decreased capacitances of the structure during bending.

The effect of humidity on the electrical resistance of the thermally rMOG is shown in Figure 9.

A commercial HIH4000 sensor was located in the humidity chamber next to the measured humidity sensors (Figure 9a). Registration of the signals from both sensors was carried out with an interval of 10 s. The measurement showed an increase in the electrical resistance of the structure, with the increase in humidity, reaching saturation at 65% and the opposite effect with the decreasing humidity (Figure 9b,c). The sensitivity of such structures, determined as ∆R/R_0_/∆RH (∆R—measured resistance range, R_0_—initial resistance, ∆RH—humidity range), was ~9%. For thermally reduced GO, at temperatures of 150 °C, the sensitivity of humidity sensors was 55% [9]. Such a large difference in the sensitivity of humidity sensors based on rGO and rMOG can be explained by the difference in the content of oxygen groups. The atomic oxygen content after thermal reduction in the case of rGO was 16%, for MOG—7%. It is known that thermal reduction at 200 °C almost completely removes hydroxyl groups. As can be seen from the XPS data in Figure 3a,b, the amount of carboxyl groups in GO and MOG is approximately the same. There is a difference in the content of epoxy groups, which is possibly the reason for the difference in the sensitivity to humidity. In general, the sensitivity of resistive humidity sensors based on GO strongly depends on the structure of the material. For GO or partially reduced GO, sensitivity varies within a wide range of 0.1–105% [20]. With a decrease in the humidity, there is a time delay in the electrical resistance change, which may be due to the accumulation of water in the interlayer spaces. A similar dependence of resistance on humidity has been observed for sensors based on thermally reduced graphene oxide [9]. The sensitivity of the MOG film to humidity can be explained by a decrease in the number of charge carriers. The reduced MOG has a p-type conductivity, similar to the reduced graphene oxide film. Water molecules interacting with the MOG film act as electron donors [21]. The relatively low sensitivity value can be associated with a rather large layer thickness; as a result, the change in surface resistance is shunted by the conductivity of the lower layers, as well as by a high degree of MOG reduction and the decrease in the number of oxygen functional groups, interacting with water molecules via Van-der-Waals forces. The transfer of electrons from water molecules to the p-type rMOG film leads to an increase in surface resistance with increasing humidity.

## 4. Conclusions

Humidity sensors and supercapacitors were obtained by screen printing of MOG suspension on a flexible PET substrate. Thermal and laser reduction was used to increase the conductivity of the MOG films. It was found that laser reduction was the best strategy for obtaining highly conductive rGO films on polymer substrates. Using a solid-state laser with a wavelength of 474 nm and a power of 200 mW, it was possible to achieve electrical resistance values of ~0.065 kΩ/sq for the MOG films with thicknesses in the range of 4–6 μm. The resistance of MOG films reduced by thermal annealing at temperatures not damaging the substrate was comparatively high. Supercapacitor structures wer obtained on a flexible substrate with a capacity of ~6 μF/cm^2^. The capacitances of such structures began to decrease from bending radii equal to ~6 mm and for a 2 mm bending radius, the capacitance decreased to 3.5 μF/cm^2^. The effect of air humidity on the resistance of humidity sensors was investigated. It was shown that their sensitivity was ~9%. Weak sensitivity was determined by a relatively large layer thickness, as a result of which the change in surface resistance was shunted by the conductivity of the lower layers, and also by a high degree of MOG reduction and the decrease in the number of oxygen functional groups, which interacts with water molecules via Van-der-Waals forces.

The screen-printed MOG structures with submicron thickness on a flexible PET substrate are promising candidates for being used in flexible electronics as electrodes for supercapacitors and humidity sensors.

## Figures and Tables

**Figure 1 materials-15-01256-f001:**

Patterns printed with a stencil on polyethylene terephthalate (PET) with dimensions: (**a**)—20.2 × 20 mm^2^; (**b**)—10.5 × 10 mm^2^; (**c**)—20 × 10.5 mm^2^; (**d**)—10.5 × 10 mm^2^; (**e**)—10 × 2 mm^2^ with a thickness of 4–6 microns.

**Figure 2 materials-15-01256-f002:**
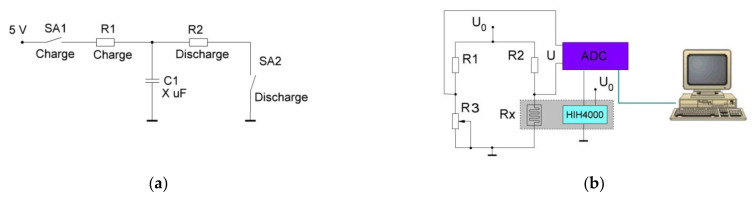
Installation diagram for testing the structures of (**a**)—supercapacitor, (**b**)—humidity sensor.

**Figure 3 materials-15-01256-f003:**
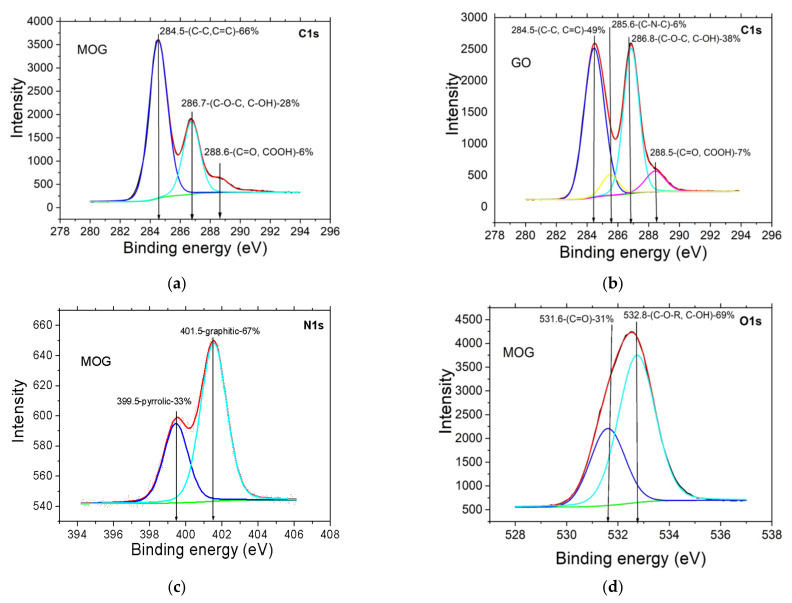
XPS spectra recorded in ranges of binding energies: (**a**) mildly oxidized graphene MOG—280–294 eV, (**b**) GO—280–294 eV (**c**) MOG—394–406 eV, (**d**) MOG—528–537 eV.

**Figure 4 materials-15-01256-f004:**
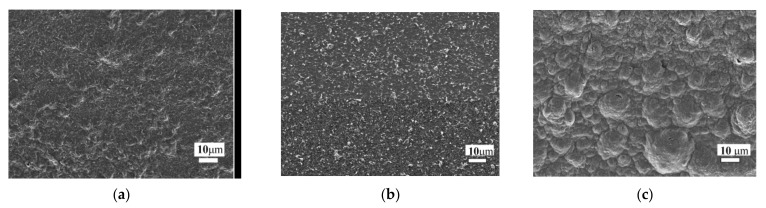
Scanning electron microscope images of MOG films before reduction (**a**), after thermal (**b**) and laser (**c**) reductions at ×1000 magnification.

**Figure 5 materials-15-01256-f005:**
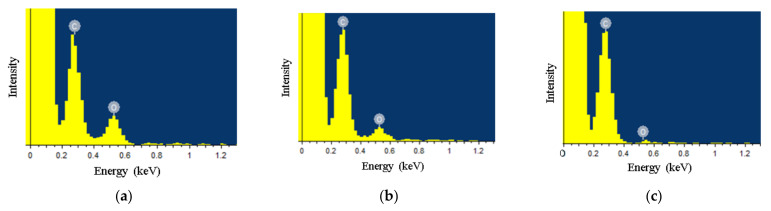
EDS spectrum for MOG (**a**)—without reduction, (**b**)—thermal reduction, (**c**)—laser reduction.

**Figure 6 materials-15-01256-f006:**
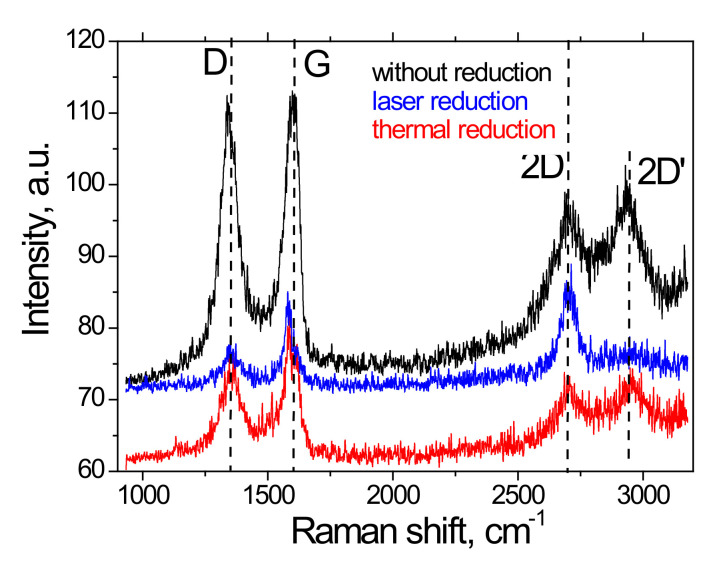
Raman spectra for the original MOG film and films reduction with a laser and thermal annealing.

**Figure 7 materials-15-01256-f007:**
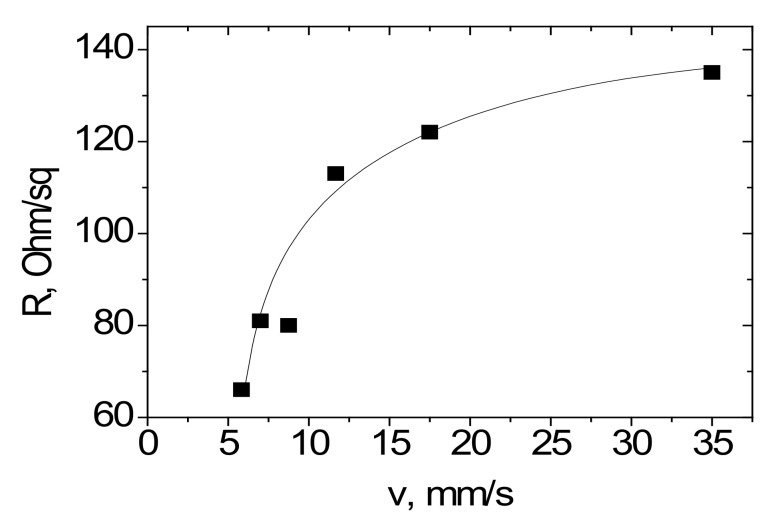
Dependence of the resistance of the MOG film on the speed of the laser beam.

**Figure 8 materials-15-01256-f008:**
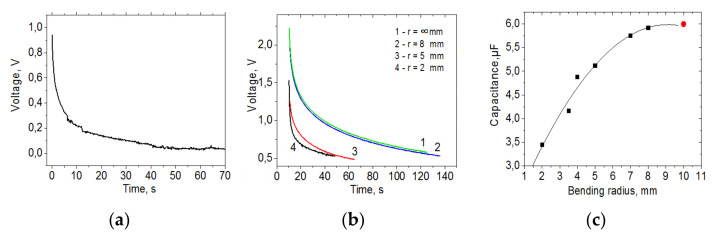
Time dependence of voltage: (**a**)—for the structure from Figure 1b; (**b**)—at different radius of deformation for the supercapacitor structure from Figure 1d; (**c**)—Dependence of the capacitance of the supercapacitor structure on the bending radius. The red dot corresponds to the capacity of the structure before bending.

**Figure 9 materials-15-01256-f009:**
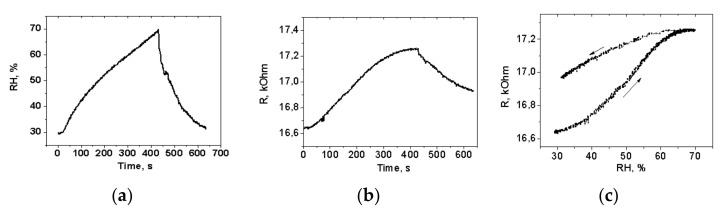
(**a**) Dependence of humidity on time measured by an commercial HIH4000 sensor (**b**) Dependence of electrical resistance on measurement time with increasing and decreasing humidity (**c**) Dependence of the resistance of the humidity sensor for a two-sided humidity sweep.

## Data Availability

Not applicable.

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
