# Peer review of "Screen-Printed Structures from a Highly Conductive Mildly Oxidized Graphene Suspension for Flexible Electronics"

_materials, 2022, doi:10.3390/ma15031256_

Round 1

Reviewer 1 Report

The reviewed manuscript concerns the results obtained for the screen-printed structures from a highly conductive mildly oxidized graphene suspension. I have the following critical remarks concerning this work:

  1. In the text, e.g. in Abstract (line 15), in Introduction (line 42), on p. 3 (+25 º), lines 116, 140, 162, 217, unit “º” is given, but I suppose that should be “ºC”.
  2. In the Introduction (lines 36-38) is written “In various studies it has been shown that GO is highly sensitive to the presence of water molecules [7-9] and is a promising material for humidity sensors.”, but for the second part of this sentence, reference should be added. On p. 2 (lines 54-55) is “The method allows a large amount of material (tens of grams [13]) to be obtained in a short period of time (10–20 min).”. Reference [13] is in the proper place? Or, for the second part of this sentence, reference should be added.
  3. On pp. 4-5 (lines 102-108) description of functional groups (e.g. epoxy (CO) and ketone (O=CO) groups) is not clear. It should be clarified.
  4. On Fig. 3b is “OG”, but should be “GO”. Axis Y should be described.
  5. For the first time, an abbreviation with the full name should be given in the text (in each of three sections: the abstract; the main text; the first figure or table), and next, only an abbreviation could be used, e.g. on p. 2 (line 63) is “MOG”, but should be “mildly oxidized graphene (MOG)”, next (line 71), should be only “MOG”. An abbreviation rMOG (e.g. on p. 5, line 130) should be defined, next (on p. 8, line 225) instead “reduced MOG”, “rMOG” should be written. On p. 2 (line 87) is “Polyethylene terephthalate”, but should be “Polyethylene terephthalate (PET)”. Also, “XPS” and “SEM”.
  6. Some units should be given with numbers without the space (e.g. %, oC), but some with the space (μF, cm). In the whole manuscript, it should be corrected, e.g. on p. 1 (line 12), should be “20-28%”, (line 20), should be “40%”, (line 21) should be “9%”, on p. 9 (line 244) should be “6 μF”, on p. 3 should be “1 cm”.
  7. On p. 2 (line 84) is “NTEGRA”, but should be “INTEGRA”.
  8. On p. 3 is “1x1cm squares were printed”, but should be “1 x1 cm squared were printed” or 1 cm x1 cm2 were printed”.
  9. On the title of Fig. 1, instead of “s”, it should be “c”, and instead of “c”, it should be “e”. The letters a, b, c, d, e should be given in parentheses (also Fig. 2). The unit (mm) is wrong. It should be “mm2” or “mm x mm”.
  10. On p. 3 is “By the method described above”, but the method is not described above.
  11. In Equation (not expression) 1, U, R1, R2, R3, Rx should be defined.
  12. On p. 5 (line 109) is “Fig. 3c”, but should be Fig. 3d” and (line 112) is “Fig. 3c”, but should be “Fig. 3d”.
  13. In Figure 5 on the x-axis, the wavenumbers are shown from the lowest to the highest values, but in spectroscopy are used from the highest to the lowest values.
  14. The figures should be improved. Text in the figures should be increased (e.g. Figs. 2, 3, 4) – is not well visible. On one figure, all graphs should be given in one size and the same position (e.g. Figs. 1, 3, 7, 8).
  15. On p. 6 (line 154) is “from 1613 cm-1 to 1592 cm-1”, but should be “from 1613 to 1592 cm-1”.
  16. On p. 8 (line 217), “ΔR/R/ΔRH” should be defined.
  17. Editorial mistakes should be corrected e.g.

- - in some places in the text, the dot should be deleted, e.g. on p. 2 (line 60), “… structure. [10].”, on. p. 3, “Fig. 1.e”, on p. 5 (line126), “works. [11].”.

- on Fig. 5, in the titles of axis’s, the space between words should be deleted, “Intensity, a.u.” and “shift, cm-1”.

  1. References should be unified according to Instructions for Authors, e.g. the name of the journal title should be given in abbreviation (e.g. in Ref. [2], “2D Mater” instead “2D Materials” should be written, in Ref [3], “Nat. Energy” instead “Nature Energy”. In References (e.g. [1-5]) is et al., but all co-authors should be given. The dot should be given at the end of Ref. [1]. The dot between the first and middle names' first letters should be deleted, e.g. Ref. [1, 9, 18].
  2. English should be carefully checked, e.g.

- on p. 1 (line 36), in “… for various sensos. In various studies …” should be “sensors”, and repeating the word “various” sounds not good.

- on p. 2 (line 59), is “… has a number of advantages. One of the most significant advantages …”, but repeating words “advantages” sounds not good,

- on p. 2 (lines 86-87), is “… using a stencil cut using …”, but repeating the word “using” sounds not good,

- on p. 5 (lines 117-118), the sentence “The surface morphology of 117 the MOG are shown in Figure 4.” is wrong.

- on p. 6 (line 163-164), the sentence “It is known that at this temperature the removal of hydroxyl and epoxy groups occurs” is wrong.

- on p. 8 (line 208-209), sentence “The effect of humidity on the electrical resistance of the thermally reduced MOG are 208 shown in Figure 8.” is wrong.

According to mentioned above remarks I suggest that in this paper the major revision is needed before publication in Materials.

Author Response

Thank you for a very detailed review of our article, as well as for pointing out the shortcomings in our study. It is great opportunity for us to receive feedback from the world scientific community, since we are geographically located in a remote area.

Reviewer 2 Report

In this manuscript, the screen-printed flexible humidity sensor and supercapacitor structures from a suspension of mildly oxidized graphene (MOG) was obtained. The structures representing a flat supercapacitor had an average specific capacitance of ~6 μF. Tensile deformations occurring during bending reduced the capacitance of supercapacitors by 40 % at a bending radius of 2 mm. Humidity sensing structures with sensitivity to air humidity of 9 % were obtained. But it seems there are no “discussion” and only “results” are shown. There should be more discussion for the advantage of this manuscript.  In this case, I suggest that the author should revise the article before the acceptance.

  1. In the abstract part, the authors mentioning “The structures representing a flat supercapacitor had an average specific capacitance of ~6 μF., …”,this is wrong, the specific capacitance should be 6 μF/cm2 or 6 μF/cm3, even 6 μF/g, the specific capacitance is connected with the volume or area or mass.
  2. In Figure 4, the scale bar in write color is too small to see.
  3. In page 5 line 115, the authors mentioning “two types of reduction were used: thermal at 200 °”, in fact, the unit of temperature should be °C. In all the manuscript, the unit of temperature is wrong.
  4. The layout in all the figures in this manuscript should be further improved.
  5. The picture quality is very poor, such as Figure 2,
  6. There are many formatting errors in the references, such as in some journal names are in full name, some of them are abbreviated, they should be uniform.; in Ref. 8, the page should not in bold type.

Author Response

Thank you for your review. Working on the problems that you identified allowed us to improve the article. It is great opportunity for us to receive feedback from the world scientific community, since we are geographically located in a remote area.

Reviewer 3 Report

At this stage, I recommend “Major Revision”. My concerns are listed below:

  • Line 76-78: “Elemental analysis based on energy dispersive X-ray spectroscopy was performed using a standard Oxford Instruments (OINA, United Kingdom) microanalytical nanoanalysis system for scanning and transmission electron microscopes.”

The authors mention EDS as well as SEM and TEM techniques in their “Materials and Methods” section. How no EDS or TEM results have been presented in their “Results and Discussion” section. Revise the manuscript for consistency

  • Line 97-98: “Determination of the type and ratio of functional groups in MOG films were made using the XPS method”

Experimental details of XPS measurements are missing. Provide relevant details

  • Line 120/Figure 4: Which mode were these images obtained?? What was the working distance?? Provide all relevant details

Author Response

Thank you for your review. We hope that the work on the shortcomings that you noted allowed us to improve our article. It is great opportunity for us to receive feedback from the world scientific community, since we are geographically located in a remote area.

Round 2

Reviewer 1 Report

The reviewed manuscript concerns the results obtained for the screen-printed structures from a highly conductive mildly oxidized graphene suspension. The most of the reviewer's comments were taken into account. I have the following critical remarks concerning this work:

  1. I’m not sure that the description of functional epoxy group as CO is correct (on p. 5, line 158). Epoxides are chemical compounds containing a three-membered cyclic group consisting of an oxygen atom and two carbon atoms, so they don’t have only CO. It should be clarified.
  2. For the first time, an abbreviation with the full name should be given in the text (in each of three sections: the abstract; the main text; the first figure or table), and next, only an abbreviation could be used, e.g. on p. 2 (line 58) is “MOGs”, next (line 60), is “mildly oxidized graphene (MOG)”.
  3. On the title of Figs. 2 and 5, the letters a, b and c should be given in parentheses.
  4. On Fig. 2, some letters in the word “discharge” are not visible. On one figure, all graphs should be given in one size and the same position (e.g. Fig. 8).
  5. Editorial mistakes should be corrected, e.g. in some places in the text, the space should be deleted, e.g. on p. 3 (line 110), is “Fig. 1 d”, but should be “Fig. 1d”, in some places in the text the space between words should be added, e.g., p. 10 (line 371), “J.Recent.”.
  6. References should be unified according to Instructions for Authors, e.g. The dot should be given at the end of Ref. [1, 8, 10, 11]. The space between the first and middle names' first letters should be deleted, e.g. Ref. [1, 2, 9, 18].

According to mentioned above remarks I suggest that in this paper the minor revision is needed before publication in Materials.

Author Response

Thank you for your review. We hope that the work on the shortcomings that you noted allowed us to improve our article. 

Reviewer 2 Report

In this manuscript, the screen-printed flexible humidity sensor and supercapacitor structures from a suspension of mildly oxidized graphene (MOG) was obtained. The structures representing a flat supercapacitor had an average specific capacitance of ~6 μF. Tensile deformations occurring during bending reduced the capacitance of supercapacitors by 40 % at a bending radius of 2 mm. Humidity sensing structures with sensitivity to air humidity of 9 % were obtained. But it seems there are no “discussion” and only “results” are shown. There should be more discussion for the advantage of this manuscript.  In this case, I suggest that the author should revise the article before the acceptance.

  1. In the abstract part, the authors mentioning “The structures representing a flat supercapacitor had an average specific capacitance of ~6 μF., …”,this is wrong, the specific capacitance should be 6 μF/cm2 or 6 μF/cm3, even 6 μF/g, the specific capacitance is connected with the volume or area or mass.
  2. In Figure 4, the scale bar in write color is too small to see.
  3. In page 5 line 115, the authors mentioning “two types of reduction were used: thermal at 200 °”, in fact, the unit of temperature should be °C. In all the manuscript, the unit of temperature is wrong.
  4. The layout in all the figures in this manuscript should be further improved.
  5. The picture quality is very poor, such as Figure 2,
  6. There are many formatting errors in the references, such as in some journal names are in full name, some of them are abbreviated, they should be uniform.; in Ref. 8, the page should not in bold type.

In this version, the manuscript was improved but still there are some problems.

  1. The layout of the Figures is still poor. Such as Figure 8 and Figure2. Some of the figures can be combined to one figure, such as Figure 6 and Figure 7. The quality of the figures is still poor, they will reduce the reputation of the journal.
  2. In Electrical characteristics of rMOG films part “an average specific capacitance of ~6 μF/cm2.” in fact, this was the lowest specific capacitance compared with the Reference, the author should describe the advantages of your work and more discussion should be added in this part.

Author Response

(The authors gave the same response as above.)

Reviewer 3 Report

Accept 

Author Response

Thank you!

Round 3

Reviewer 2 Report

it should be accepted